# MiR-155-5p and MiR-203a-3p Are Prognostic Factors in Soft Tissue Sarcoma

**DOI:** 10.3390/cancers12082254

**Published:** 2020-08-12

**Authors:** Thomas Greither, Franziska Koser, Hans-Jürgen Holzhausen, Antje Güttler, Peter Würl, Matthias Kappler, Sven Wach, Helge Taubert

**Affiliations:** 1Center for Reproductive Medicine and Andrology, Martin Luther University Halle-Wittenberg, 06120 Halle (Saale), Germany; 2Institute of Physiology II, University of Muenster, 48149 Münster, Germany; Franziska.koser@ukmuenster.de; 3Institute of Pathology, Martin Luther University Halle-Wittenberg, 06120 Halle (Saale), Germany; Hans-juergen.holzhausen@medizin.uni-halle.de; 4Department of Radiotherapy, Martin Luther University Halle-Wittenberg, 06120 Halle (Saale), Germany; antje.hahnel@medizin.uni-halle.de; 5Department of General and Visceral Surgery, Hospital Dessau, 06847 Dessau-Roßlau, Germany; peter.wuerl@klinikum-dessau.de; 6Department of Oral and Maxillofacial Plastic Surgery, Martin Luther University Halle-Wittenberg, 06120 Halle (Saale), Germany; matthias.kappler@uk-halle.de; 7Clinic of Urology and Pediatric Urology, FA University Hospital Erlangen-Nürnberg, 91054 Erlangen, Germany; sven.wach@uk-erlangen.de (S.W.); helge.taubert@uk-erlangen.de (H.T.)

**Keywords:** soft tissue sarcoma, prognosis, miR-155-5p, miR-203a-3p

## Abstract

Soft tissue sarcoma (STS) is a heterogeneous group of rare malignancies with a five-year survival rate of approximately 50%. Reliable molecular markers for risk stratification and subsequent therapy management are still needed. Therefore, we analyzed the prognostic potential of miR-155-5p and miR-203a-3p expression in a cohort of 79 STS patients. MiR-155-5p and miR-203a-3p expression was measured from tumor total RNA by qPCR and correlated with the demographic, clinicopathological, and prognostic data of the patients. Elevated miR-155-5p expression was significantly associated with increased tumor stage and hypoxia-associated mRNA/protein expression. High miR-155-5p expression and low miR-203a-3p expression, as well as a combination of high miR-155-5p and low miR-203a-3p expression, were significantly associated with poor disease-specific survival in STS patients in the Kaplan–Meier survival analyses (*p* = 0.027, *p* = 0.001 and *p* = 0.0003, respectively) and in the univariate Cox regression analyses (RR = 1.96; *p* = 0.031; RR = 2.59; *p* = 0.002 and RR = 4.76; *p* = 0.001, respectively), but not in the multivariate Cox regression analyses. In conclusion, the oncomiR miR-155-5p and the tumor suppressor-miR miR-203a-3p exhibit an association with STS patient prognosis and are suggested as candidates for risk assessment.

## 1. Introduction

Soft tissue sarcoma (STS) comprises approximately 50 subtypes of tumors characterized by their mesenchymal origin and heterogeneous phenotype [1]. Although STS is a rare tumor entity, with an incidence rate of only 3.6 per 100,000 [2], the low five-year survival rate of 50% is a major issue [3]. Therefore, there is still an urgent need for reliable, specific, and easily measurable prognostic markers for risk stratification and subsequent therapy management.

MicroRNAs are 18–25 nucleotide long, noncoding RNAs that post-transcriptionally regulate gene expression by repressing the translation of their respective target genes through RNA-induced silencing complex (RISC)-mediated binding to the 3′-UTR and subsequent inhibition of proper ribosomal assembly [4]. Frequently, microRNA expression patterns are specific for physiologic tissues, but also systemic diseases or malignancies. In this context, sarcomas have been subject to a characterization of the individual microRNA patterns putatively associated with their heterogeneous appearance and tumor-biological behavior (reviewed in [5]).

Furthermore, there is a multitude of studies showing a considerable impact of the dysregulation of certain microRNAs in all major malignancies on the prognosis of patients. Previously, we were able to demonstrate an association between increased miR-155, miR-210, or miR-203 expression and a significantly worsened prognosis in pancreatic adenocarcinoma [6]. An intermediate miR-210 level was also associated with worsened survival in STS patients [7].

MiR-155-5p (henceforth designated miR-155) and its complementary strand miR-155-3p are processed from the B-cell integration cluster (BIC) gene located on chromosome 21, which is alternatively designated the MIR155 host gene (MIR-155HG). Although miR-155 plays a role in hematopoiesis, inflammation, and immune responses, it may act as an oncogenic miRNA. It is not only upregulated in different leukemias and lymphomas, but also in solid tumors such as cancer of the breast, colon, lung, pancreas, and thyroid (reviewed in [8]). Furthermore, miR-155 was reported to be overexpressed in liposarcoma (LS)-derived cell lines or liposarcoma biopsies in comparison to normal fat tissue [9,10] and contribute to LS progression by targeting the central Wnt pathway component casein kinase 1α (CK1α) [9]. Interestingly, the Kaposi sarcoma-associated herpesvirus genome encodes an ortholog of miR-155 [11], and overexpression of this viral-encoded microRNA is associated with the suppression of cell growth control [12]. MiR-155 is also overexpressed in osteosarcoma samples in comparison to corresponding normal tissue and targets the Wnt pathway repressor HMG-box transcription factor 1 (HBP1), thereby driving osteosarcoma cell proliferation by enhancing Wnt pathway action [13]. Further important cellular pathways regulated by miR-155 in osteosarcoma include the PTEN/AKT/mTOR pathway [14] and the NFκB pathway [15].

The miR-203a gene is located on chromosome 14q32.33 [16]. MiR-203a-3p (henceforth designated as miR-203) is expressed aberrantly in different cancers, such as gastric, pancreatic, colon and esophageal cancer, compared to normal tissue (reviewed in [17,18]), which suggests its expression to serve as a diagnostic marker. MiR-203 was identified to be downregulated both in rhabdomyosarcoma (RMS) cell lines and in rhabdomyosarcoma patient samples due to promoter hypermethylation. Re-induction of miR-203 expression by demethylating agents downregulates p63 and leukemia inhibiting factor receptor (LIFR) [19]. Furthermore, miR-203 is significantly downregulated in osteosarcoma samples and cell lines compared to controls, while its target gene RAS-related protein Rab22A (RAB22A) is overexpressed. Conversely, the overexpression of miR-203 blocks osteosarcoma growth and migration [20].

In this study, we aimed to analyze the impact of miR-155 and miR-203 on the diagnostics as well as the correlation with clinical characteristics and with the prognosis of STS patients.

## 2. Results

### 2.1. miR-155 and miR-203 Expression in STS Specimens

MiR-155 and miR-203 were detectable in all analyzed STS specimens, with median expression values of 16.7 and 1.9, respectively. The distribution of miR-155 expression values ranged from 0.3–189.5, while miR-203 expression values ranged from 0.003 to 14358.7.

By using recursive partitioning methods, we established the influence and optimized cut-offs of each microRNA when analyzing the combinational effect on STS patient survival. With an expression of miR-203 ≥ 0.56 and miR-155 ≥ 42.92, there was a 68% correct classification rate of tumor-related death in the STS patients (see Figure 1).

Optimized cut-off values were also defined by assessing the value with the highest Youden index in receiver operating characteristics (ROC) analyses, testing the sensitivity and specificity of each microRNA in the tumor-related death of the patients (see Figure 2).

The optimal cut-off was 18.0 for miR-155 and 0.9 for miR-203. When assessing low and high expression of the microRNAs with respect to age, gender, tumor stage, resection type, tumor localization, and histological subtypes of the respective tumor, we detected a significant association between increased miR-155 expression and increased tumor stage (*p* = 0.01, Kruskal–Wallis test, see Figure 3).

Regarding the histological subtype, especially liposarcoma and neuronal sarcoma exhibited a lower miR-155 expression in comparison to the other sarcoma entities (see Appendix A). These differences were significant (*p* = 0.001, Kruskal–Wallis test). However, this was based on relatively low case numbers (*n* = 7–21 per category).

### 2.2. Bivariate Correlation Analyses

We analyzed the molecular biological factors associated with miR-155 and miR-203 expression in soft tissue sarcoma. When considering all significant correlations with r_s_ > 0.25 or r_s_ < −0.25, we retrieved eight and five factors associated with miR-155 and miR-203, respectively (see Table 1).

Both miR-155 and miR-203 expression levels were significantly correlated with miR-210 expression. Moreover, miR-155 expression was significantly correlated with the expression of proteins of the urokinase-type plasminogen activator (uPA) system (r_s_ = 0.347–0.604, Spearman-Rho) and the tumor tissue level of osteopontin (OPN) in the tumors of STS patients (r_s_ = 0.361). Additionally, miR-155 expression was significantly associated with ephrin-A3 (EFNA3) mRNA expression (r_s_ = −0.378) and LGR5 mRNA expression (r_s_ = −0.389). MiR-203 was significantly inversely associated with OPN mRNA expression (splice variants a-c: r_s_ = −0.26–−0.324) and pAKT tumor tissue level (r_s_ = −0.434).

### 2.3. Survival Analyses

Finally, we tested the association of miR-155 and miR-203 expression with the prognosis of STS patients. In Kaplan–Meier survival analyses, elevated miR-155 expression (≥18.0) was significantly associated with worsened tumor-related survival (median survival time 25 months in comparison to 118 months; see Figure 4a). On the other hand, lower miR-203 expression (<0.9) was significantly associated with worsened tumor-related survival in the STS patients (median survival time 18 months vs. 86 months, see Figure 4b).

Furthermore, when combining the expression of both microRNAs, the unfavorable combination miR-155 high + miR-203 low was significantly associated with a worsened survival of the patients in comparison to the miR-155 low + miR-203 high combination in tumor tissues (*p* < 0.001, log-rank-test, see Figure 4c).

In the univariate Cox regression analyses, increased miR-155 expression (*p* = 0.03; RR = 1.96) and decreased miR-203 expression (*p* = 0.002; RR = 2.59) were also associated with a higher risk of tumor-related death (see Table 2). When combining both microRNAs, the less favorable combination miR-155 high + miR-203 low exhibited a significant 4.76-fold increased risk of tumor-related death (*p* = 0.001). However, although the estimated risk ratios remained numerically equivalent in multivariate Cox regression analyses adjusted for tumor stage, tumor localization, resection type, and STS sub-entity, the association between miR-155 or the combination of miR-155 and miR-203 was no longer an independent predictor of STS patient survival, probably because of the interdependence of increased miR-155 expression and a higher tumor stage. Furthermore, the relapse-free survival of the patients was not associated with miR-155 or miR-203 expression in uni- or multivariate Cox regression analyses (see Table 2).

.

Furthermore, when analyzing the effect of miR-155 or miR-203 on the patients survival in distinct histological subtypes, it became evident that in myogenic sarcomas (rhabdo- and leiomyosarcoma) a lowered miR-155 expression as well as a lowered miR-203 expression was significantly associated to a worsened survival of the STS patients (See Appendix A). 

## 3. Discussion

We measured miR-155 and miR-203 expression in 79 soft tissue sarcoma (STS) specimens. Both microRNAs were detectable in all samples to a different extent, with higher expression of miR-155 than miR-203. This is consistent with the general theory of miR-155 being an oncomiR (cancer-associated microRNA) defined by its general upregulation in lymphomas, leukemias, and solid tumors [8,27,28], and miR-203 being a tumor suppressor miR due to its general downregulation in many tumor entities [20,29,30]. Interestingly, several groups report miR-155 to be overexpressed, especially in liposarcoma in comparison to normal fat tissue [9,10]. We did not observe any association between the STS sub-entities present in our cohort and altered miR-155 expression; however, we could not monitor normal tissue corresponding to our tumor specimens. On the other hand, increased miR-155 expression was significantly associated with elevated tumor stage in our STS cohort. Higher expression of miR-155 in connection with advanced tumor stages was also observed in breast cancer [31], laryngeal squamous cell carcinoma [32], or chordoma [33]. In contrast, miR-155 expression in pediatric Wilms tumors was lower in advanced tumor stages and increased due to chemotherapy [34].

MiR-155 and miR-203 expression was correlated with several–mainly hypoxia-associated–molecular factors in our STS cohort. MiR-155 was significantly positively correlated with the hypoxia-dependent microRNA miR-210 and inversely correlated with the mRNA expression of the known miR-210 target gene ephrin-A3 (EFNA3) [6,35]. There is a negative feed-back-loop between miR-155 and von-Hippel-Lindau factor (VHL) [36]. On one hand, the VHL protein suppresses miR-155 and miR-210 expression. On the other hand, Kong and colleagues demonstrated the pivotal role of miR-155 in tumor angiogenesis of breast cancer tissue through targeting VHL and subsequent HIF-1α stabilization [36]. They suggest that miR-155 regulates the VHL/HIF pathway to induce tumor angiogenesis and metastasis in triple-negative breast cancer. Furthermore, miR-210 is upregulated by HIF-1α and therefore it is possible that miR-155 indirectly increases miR-210 by stabilizing hypoxia inducible factor 1 subunit alpha (HIF1α). In this way, increasing miR-210 levels can reduce mRNA levels of its target gene EFNA3.

Additionally, we found a correlation between miR-155 expression and p53 mutation status. This is in accordance with previous findings demonstrating that miR-155 can be downregulated by wild-type p53, p63 and breast cancer type 1 susceptibility protein (BRCA1). However, an upregulation of miR-155 through mutant p53 or mutant BRCA1 has been reported (reviewed in [8,37]).

Moreover, we detected a significant association between increased miR-155 expression and several components of the uPA system, namely, uPA, uPAR, and PAI-1. The expression of miR-155 is also positively correlated with the expression of serpine1/PAI-1 in breast cancer [37]. Wild-type p53 represses uPA transcription by repressing the uPA promoter and enhancer [38]. However, oncogenic p53 mutants exhibit a significant loss or total ablation of this repressing activity [38]. Altogether, the correlation of miR-155 expression with the whole uPA system warrants further investigation.

In addition, the miR-155 level is negatively correlated with the mRNA expression level of the stem cell marker leucine rich repeat containing G protein-coupled receptor 5 (LGR5) [24]. However, LGR5 does not appear to be a direct target of miR-155 (http://mirtar.mbc.nctu.edu.tw/human/). Interestingly, LGR5-induced p53 degradation was described for hepatocellular carcinoma cells, which was also associated with therapeutic resistance against doxorubicin [39]. It appears that there is a LGR5/p53/miR-155 axis, but this needs further investigation.

MiR-203 is negatively associated with the expression of several osteopontin splice variants. Interestingly, neither the existing literature nor available databases (TargetScan.org) hint at the direct targeting of OPN by miR-203. However, a reduction in OPN mRNA/protein under hypoxia has been reported in normal human cells [40]. Therefore, it remains to be elucidated whether there is a direct or indirect regulation of this gene by miR-203. Furthermore, a negative correlation between miR-203 expression and the protein level of phosphorylated AKT (pAKT) was observed in our study. This is in agreement with the findings of Yang and coworkers [41]. They identified phosphatidylinositol 4,5-bisphosphate 3-kinase (PIK3CA) as a direct target gene of miR-203. Inhibition of miR-203 resulted in increased levels of pAKT and pPIK3CA. Therefore, they suggest that miR-203 targets the PI3K/Akt signaling pathway [41].

In our cohort, increased miR-155 expression exhibited a significant negative impact on STS patient survival, which was demonstrated for different STS entities for the first time. In this way, our results support the findings of Kapodistrias and colleagues, who showed that increased miR-155 expression is an independent indicator of unfavorable prognosis in liposarcoma [42]. In non-small-cell lung cell carcinoma, the effect of miR-155 expression on patient survival was dependent on the histological subtype, with negative effects of increased miR-155 expression in adenocarcinoma and decreased miR-155 expression in squamous cell carcinoma [43]. However, in general, increased miR-155 expression is associated with worsened survival in tumor patients, as seen in pancreatic ductal adenocarcinoma [6], colorectal carcinoma [44], hepatocellular carcinoma [45], breast cancer [46], glioma [47], osteosarcoma [13], or leukemia [48].

On the other hand, we identified, for the first time in STS, a significant association between a lower miR-203 expression and a worsened STS patient outcome. The role of miR-203 in tumor patient survival is multifaceted, with increased miR-203 being a negative predictor of patient survival in pancreatic adenocarcinoma [6,49], epithelial ovarian cancer [50], breast cancer [51], colon/colorectal cancer [52,53], and ependymoma [54], but being a positive predictor for patient survival in glioma [55], hepatocellular carcinoma [56], epithelial ovarian cancer [57], and esophageal cancer [17]. MiR-203 has been characterized to act as a tumor suppressor microRNA in laryngeal carcinoma by targeting survivin [58] and in lung cancer by targeting v-src avian sarcoma viral oncogene (SRC) and protein kinase C alpha (PKCα/PRKCA) [57,59]. Furthermore, a tumor suppressive role of miR-203 was shown in prostate cancer [60]. MiR-203 levels are downregulated in clinical samples of primary prostate cancer and further reduced in metastatic prostate cancer [60]. Transfection with miR-203 precursors resulted in a reduction of phosphorylated EGFR and phosphorylated ERK1/2 in DU145 derivative PCa cells. Siu et al suggested that the loss of miR-203 is a molecular link in the progression of prostate cancer metastasis and tyrosine kinase inhibitor (TKI) resistance characterized by high EGFR ligand output and antiapoptotic protein activation [60]. Interestingly, hypoxia can upregulate phosphorylated ERK1/2 in normal human cells [40], and a reduction in miR-203 level is associated with increased phosphorylated ERK1/2 as well [60]. Therefore, both hypoxia and a reduction in miR-203 level may trigger activation of MAPK (ERK1/2)/AKT signaling, which also plays a role in different sarcomas [61].

Altogether, increased miR-155 is associated with more aggressive behavior and poor prognosis in many different cancers, but for miR-203, a controversial picture can be drawn depending on the tumor entity. In this way, miR-155 appears to be a potential target for miRNA-based therapy, as suggested seven years ago [8]. However, the physiological role of miR-155 in myelopoiesis and erythropoiesis should also be considered [8]. Since that time, miRNA therapies have undergone evaluation in several recently undertaken clinical trials where results are eagerly awaited [62]. MiR-155 inhibition by MRG106/cobomarsen was recently tested in a clinical phase 1 study in different leukemia and lymphoma patients (NCT02580552) and in two clinical phase 2 studies in cutaneous T-cell lymphoma/mycosis fungoides patients (NCT03837457, NCT03713320). Our results suggest that inhibition of miR-155 could also have potential as a future treatment option for STS patients.

## 4. Materials and Methods

### 4.1. Patients and Specimens

The tumor specimens analyzed in this study originated from 79 adult patients from a previously described STS cohort [63]. The study was approved by the ethics committee of Martin Luther University Halle-Wittenberg (date of approval: 24 January 2007). All patients gave written informed consent. The demographic and clinicopathological details of the patients are summarized in Table 3.

### 4.2. RNA Isolation and cDNA Synthesis

Total RNA was isolated by phenol/chloroform extraction using TRIzol reagent (Invitrogen, Karlsruhe, Germany) from 10–30 tumor slides of 10 μm thickness. Briefly, tumor tissue was suspended in TRIzol reagent and vortexed vigorously, and then chloroform (Roth, Karlsruhe, Germany) was added and mixed. The suspension was centrifuged for 10 min at 6000× *g*, and the aqueous phase was mixed with isopropanol (Roth). After precipitation overnight, the RNA pellet was washed twice in ice-cold ethanol (96% and 70%, respectively; Merck, Darmstadt, Germany) and eluted in 20 μL DEPC-treated water (Invitrogen).

Ten nanograms of total RNA was introduced to cDNA synthesis of miR-155, miR-203 and U18 snoRNA (reference gene) with the stem-loop primer system (Applied Biosystems, Darmstadt, Germany). Reverse transcription reactions were carried out with the SuperScript II Reverse transcription kit (Invitrogen, Karlsruhe, Germany) according to the manufacturers’ protocols. The qPCR measurements were performed in duplicate with the HotStart Taq Polymerase Kit (Qiagen, Hilden, Germany) on the Rotorgene 3000 real-time PCR system (LTF, Wasserburg, Germany). The resulting mean C_T_ values were analyzed in comparison to U18 snoRNA expression as a reference gene. Further, 2^−ΔCT^ values were designated according to Livak and Schmittgen [64].

### 4.3. Statistical Analysis

Bivariate linear regression analysis (Spearman’s rank test) was applied to test the correlation between miR-155 or miR-203 expression and molecular biological tumor markers. Pearson’s Chi^2^ test and nonparametric tests (Kruskal-Wallis test) were applied to detect interdependences between microRNA expression and demographic or clinicopathological parameters. Survival analyses (Kaplan–Meier analyses, uni- and multivariate Cox regression analyses) were performed to test associations between microRNA expression and STS patient survival. All these tests were performed with SPSS 25.0 (IBM, Ehingen, Germany). *p* values ≤ 0.05 were considered significant. ROC analyses and recursive partitioning were applied to test the individual contributions of miR-155 and miR-203 in combined risk stratification. These tests were performed with the R statistical framework (v. 3.2.1; R foundation for statistical computing, Vienna, Austria) and the additional function libraries pROC and rpart.

## 5. Conclusions

We analyzed the association of miR-155 and miR-203 expression with the prognosis of soft tissue sarcoma patients. Both microRNAs were detectable in all tumor specimens to different extents and exhibited correlations with several hypoxia-associated molecular markers, such as miR-210 and, in the case of miR-155, components of the urokinase-like plasminogen activator (uPA) system and ephrin-A3 (EFNA3). Furthermore, miR-155 expression was higher in advanced tumor stages. The expression of miR-155 and miR-203 as well as the combination of both microRNAs was significantly associated with a worsened disease-specific survival in soft tissue sarcoma patients. In summary, miR-155 and miR-203 are interesting candidates for risk stratification in soft tissue sarcoma patients, and miR-155 may be an option for therapy management in the future.

## Figures and Tables

**Figure 1 cancers-12-02254-f001:**
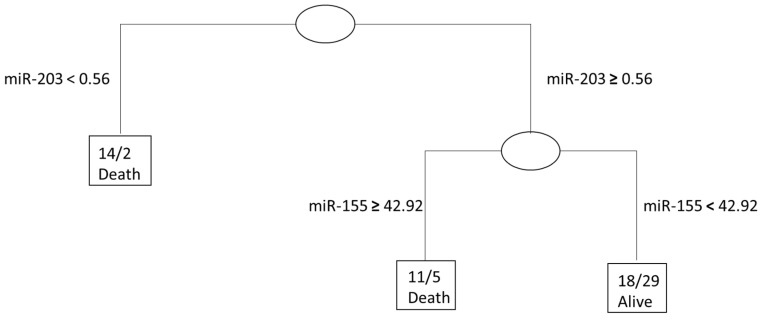
Classification tree establishing the influence of each microRNA in a combinational approach towards risk stratification.

**Figure 2 cancers-12-02254-f002:**
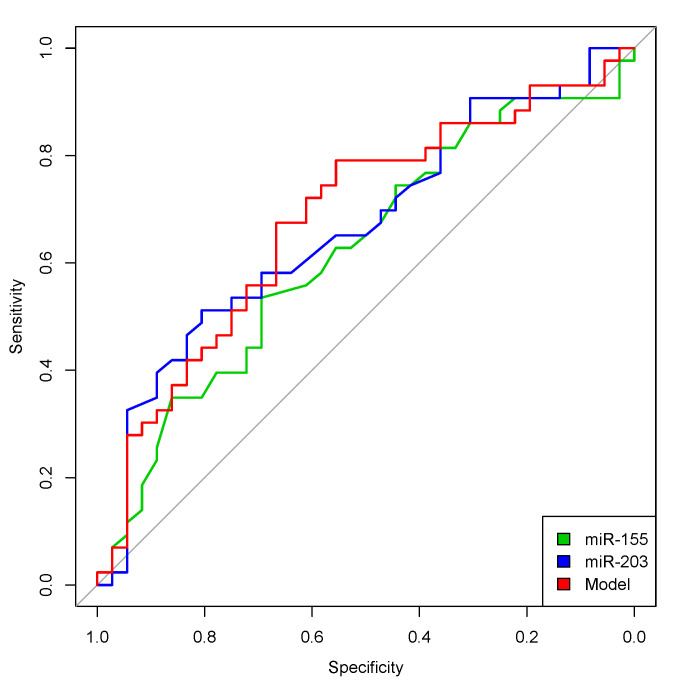
ROC analyses for miR-155 (green), miR-203 (blue) and a combination of miR-155 + miR-203 (model in red) and their prognostic application. Both microRNAs exhibited fair area under the curve values (AUCs; miR-155: AUC 0.62 (0.49–0.74), *p* = 0.067; miR-203: AUC 0.67 (0.55–0.79), *p* = 0.011). MiR-155 or miR-203 expression with the highest Youden index according to single ROC analyses was chosen as the cut-off for subsequent survival analyses. A fitted linear regression model that combined miR-155 and miR-203 also exhibited only a fair AUC of 0.68 (0.56–0.80), *p* = 0.006.

**Figure 3 cancers-12-02254-f003:**
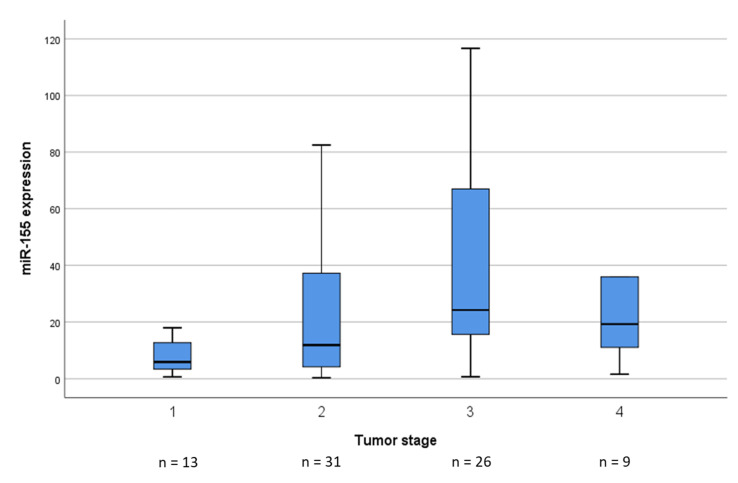
The distribution of miR-155 expression in relation to tumor stage. MiR-155 expression was significantly associated with tumor stage (*p* = 0.01, Kruskal-Wallis test).

**Figure 4 cancers-12-02254-f004:**
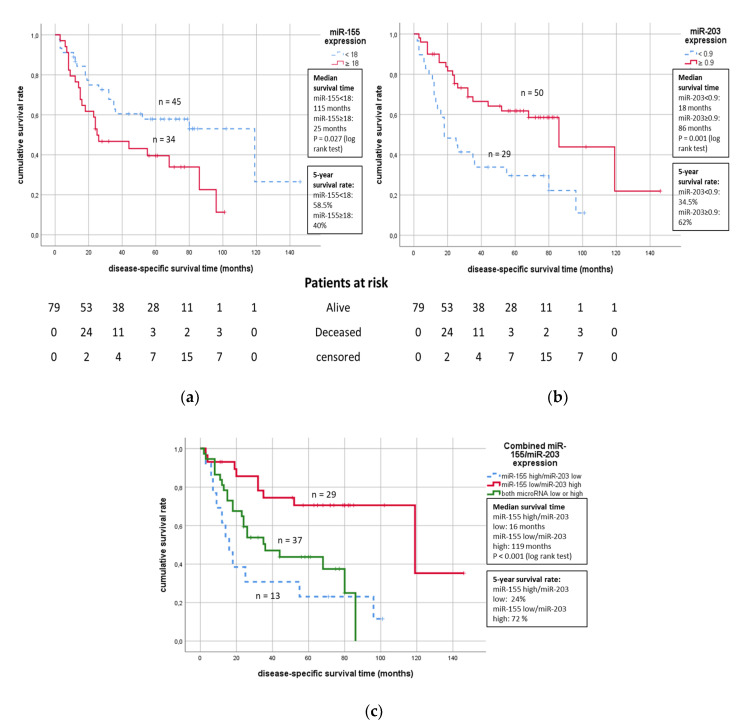
Kaplan-Meier survival analyses of the prognostic impact of miR-155 (**a**) and miR-203 (**b**) expression or a combination of both microRNAs (**c**) on soft tissue sarcoma patient survival. Below the figures, the number of patients at risk at specific time points is given.

**Table 1 cancers-12-02254-t001:** Bivariate correlation analyses (Spearman-Rho) with miR-155 or miR-203 and different molecular-pathological factors in soft tissue sarcoma.

Parameters		r_s_	*p*	*n*	Reference
miR-155	p53 mutations	0.260	0.040	63	[21]
	OPN protein	0.361	0.002	72	[22]
	uPA protein	0.604	<0.001	79	[23]
	uPAR protein	0.440	<0.001	79	[23]
	PAI-1 protein	0.347	0.002	79	[23]
	EFNA3 mRNA	−0.378	0.001	75	*n.p.*
	LGR5 mRNA	−0.389	0.001	69	[24]
	miR-210	0.349	0.002	78	[7]
miR-203	miR-210	0.265	0.019	78	[7]
	OPN-a mRNA	−0.267	0.021	74	[25]
	OPN-b mRNA	−0.324	0.005	74	[25]
	OPN-c mRNA	−0.260	0.025	74	[25]
	pAKT473 protein	−0.434	<0.001	73	[26]

Abbreviations: r_s_—correlation factor according to Spearman-Rho; *n.p.*—not published own data.

**Table 2 cancers-12-02254-t002:** Univariate and multivariate Cox regression survival analyses of the impact of miR-155 and miR-203 on soft tissue sarcoma patient survival. Multivariate Cox regression analyses adjusted for tumor stage, resection type, tumor localization and tumor sub-entity. RR = relative risk with 95% confidence interval in brackets.

Parameters	Univariate Cox Regression Analysis			Multivariate Cox Regression Analysis			
	*n*	*p*	RR	*n*	*p*	RR	
miR-155 low	45		Reference	45		Reference	**Tumor-specific survival**
miR-155 high	34	**0.031**	**1.96 (1.06–3.61)**	34	0.766	1.13 (0.52–2.46)	
miR-203 low	29	**0.002**	**2.59 (1.41–4.76)**	29	0.077	1.99 (0.93–4.25)	
miR-203 high	50		Reference	50		Reference	
miR-155 low + miR-203 high	29		Reference	29		Reference	
miR-155 high + miR-203 low	13	**0.001**	**4.76 (1.89–12.05)**	13	0.21	4.05 (0.46–35.7)	
miR-155 low	45		Reference	45		Reference	**Relapse-free survival**
miR-155 high	34	0.985	1.01 (0.5–2.01)	34	0.512	1.32 (0.34–1.73)	
miR-203 low	29	0.189	1.61 (0.79–3.26)	29	0.405	1.60 (0.21–1.90)	
miR-203 high	50		Reference	50		Reference	
miR-155 low + miR-203 high	29	0.455	1.5 (0.23–1.93)	29		Reference	
miR-155 high + miR-203 low	13		Reference	13	0.247	3.04 (0.46–19.99)	

Significant values are in bold face.

**Table 3 cancers-12-02254-t003:** MiR-155 and miR-203 expression in relation to clinical and histopathological characteristics in soft tissue sarcoma patients.

Parameters		miR-155 Low (<18.0)	miR-155 High (≥18.0)	Chi^2^ Test (*p*-Value)	miR-203 Low (<0.9)	miR-203 High (≥0.9)	Chi^2^ Test (*p*-Value)
age	<60 years	23	20	n.s.	15	28	n.s.
	>60 years	22	14		14	22	
gender	female	24	12	n.s.	16	20	n.s.
	male	21	22		13	30	
patients status	alive	25	11	0.04	7	29	0.004
	deceased	20	23		22	21	
tumor stage ^a^	I	12	1	0.011	3	10	n.s.
	II	19	12		10	21	
	III	11	15		11	15	
	IV	3	6		5	4	
resection	radical (R0)	32	21	n.s.	17	36	n.s.
	not radical (R1)	13	13		12	14	
tumor localization	extremities	29	23	n.s.	18	34	n.s.
	trunk wall	3	2		2	3	
	head/neck	2	0		1	1	
	abdomen/peritoneum	9	9		6	12	
	multiple locations	2	0		2	0	
histological subtypes	LS	12	9	0.011	10	11	n.s.
	FS + NOS	12	8		4	16	
	NS	4	3		3	4	
	RMS + LMS	11	10		8	13	
	other	6	4		4	6	
tumor size	T1	6	5	n.s.	3	8	n.s.
	T2	39	29		26	42	
number of relapses	0	28	20	n.s.	15	33	n.s.
	1	8	5		6	7	
	>2	9	9		8	10	

^a^ Union for International Cancer Control Guidelines; Abbreviations: LS—liposarcoma; FS—fibrosarcoma; RMS—rhabdomyosarcoma; LMS—leiomyosarcoma; NS—neuronal sarcoma; Syn—synovial sarcoma; NOS—not otherwise specified; T1—≤5 cm in diameter; T2—>5 cm in diameter; n.s.—not significant.

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
