# Peer review of "MiR-155-5p and MiR-203a-3p Are Prognostic Factors in Soft Tissue Sarcoma"

_cancers, 2020, doi:10.3390/cancers12082254_

Round 1

Reviewer 1 Report

The manuscript describes a study of soft tissue sarcoma patients for the role of two miRNAs in disease progression.

The major concern is the lack of a proper control, i.e.e normal healthy samples to find out the baseline levels of these miRNAs. all analysis is based on defat Ct values while if they include a norma healthy control, they can get Delta delta ct and the findings will have more clinical value.

The current writing fo discussion gives the impression that this study is not finding anything new but simply confirming what is already known in the literature from other studies.

Minor comments: There is a need for checking formatting and sentence formation at multiple place, I am citing a few examples here:

  1. Line 263-264: "Total RNA was isolated by phenol/chloroform extraction using TRIzol reagent (Invitrogen, 263 Karlsruhe, Germany) from 10 – 30 tumor slides of 10 m thickness" I assume they missed the correct unit as 10 m is not right!
  2. Line 230-231: "kinase C alpha (PKCα/PRKCA) [52,54]. Furthermore, a tumor suppressive role of miR-203 was shown 

    In prostate cancer" There is change of paragraph on a run off sentence.

  3. Line 82 -83: "In this study, we aimed to analyze the impact of miR-155 and miR-203 on the diagnostic and clinical characteristics and prognosis of soft tissue sarcoma patients". There is too much use of "and". 

  4. Line 60-62: "It is upregulated in different leukemias and lymphomas but also in solid tumors such as cancer of the breast, colon, lung, pancreas and thyroids (reviewed in [8])." it is grammatically wrong, it should be written as  "It is not only upregulated in different leukemias and lymphomas but also in solid tumors such as cancer of the breast, colon, lung, pancreas and thyroids (reviewed in [8]).

Author Response

The manuscript describes a study of soft tissue sarcoma patients for the role of two miRNAs in disease progression.

Point 1: The major concern is the lack of a proper control, i.e.e normal healthy samples to find out the baseline levels of these miRNAs. all analysis is based on defat Ct values while if they include a norma healthy control, they can get Delta delta ct and the findings will have more clinical value.

Response 1: We agree with the reviewer that the inclusion of miR-155 and miR-203 expression data in corresponding normal tissue would have added further value to the study. Unfortunately, these specimens were not available for the analysis in our cohort. Therefore, the expression data could only be displayed as Delta CT values.

Point 2: The current writing fo discussion gives the impression that this study is not finding anything new but simply confirming what is already known in the literature from other studies.

Response 2: Indeed, miR-155 and miR-203 are relatively well described as oncoMiR or tumorsuppressor MiR, respectively, in a wide range of tumor entities. However, for soft tissue sarcomas, an association of these microRNAs with the prognosis of the patients have not – to the best of our knowledge – shown before. We tried to indicate this novelty in l. 232 and l. 241; however without overstating our results. Especially interesting – to our opinion – is the impact of miR-203 on the STS patients survival, as increased miR-203 may be a double-edged sword, being a negative predictor of patient survival in pancreatic adenocarcinoma, epithelial ovarian cancer, breast cancer, colon/colorectal cancer and ependymoma but being a positive predictor for patient survival in glioma, hepatocellular carcinoma, epithelial ovarian cancer and esophageal cancer (l. 242 – 246). In the case of STS, according to our results, it seems that the tumor suppressor-MiR function is predominant. To our opinion, this is an important finding warrant further mechanistic studies on the function of miR-203 in STS development and progression.

Point 3: Minor comments: There is a need for checking formatting and sentence formation at multiple place, I am citing a few examples here:

  1. Line 263-264: "Total RNA was isolated by phenol/chloroform extraction using TRIzol reagent (Invitrogen, 263 Karlsruhe, Germany) from 10 – 30 tumor slides of 10 m thickness" I assume they missed the correct unit as 10 m is not right!
  2. Line 230-231: "kinase C alpha (PKCα/PRKCA) [52,54]. Furthermore, a tumor suppressive role of miR-203 was shown 

In prostate cancer" There is change of paragraph on a run off sentence.

  1. Line 82 -83: "In this study, we aimed to analyze the impact of miR-155 and miR-203 on the diagnostic and clinical characteristics and prognosis of soft tissue sarcoma patients". There is too much use of "and". 
  2. Line 60-62: "It is upregulated in different leukemias and lymphomas but also in solid tumors such as cancer of the breast, colon, lung, pancreas and thyroids (reviewed in [8])." it is grammatically wrong, it should be written as  "It is not only upregulated in different leukemias and lymphomas but also in solid tumors such as cancer of the breast, colon, lung, pancreas and thyroids (reviewed in [8]).

Response 3: We applied the corrections at the respective places (marked in yellow).

Finally, we thank the reviewer for the thorough review of our manuscript and the effort of pointing out parts of the manuscript which needed clarification. We hope we applied the changes and additional analyses to his or her satisfaction. Best regards. Thomas Greither (in behalf of the authors)

Reviewer 2 Report

In the manuscript, the authors report the value of miR155 and miR203 expression in soft tissue sarcoma (STS) cohort including some histo-subtypes of total 79 tumors for their prognostic potential.

The study supports the possibility of miR155 working as an oncomiR as known in some cancer types including liposarcoma, and of miR203 as a tumor suppressor.

The concern in the study design is that the cohort includes mixed histotypes and each with only a small number so they end up treating STS as a single disease entity.

STS is a large entity of neoplasm of mesenchymal origin including more than 50 different histo-types.

Carcinoma is a large category including all neoplasm of epithelial origin, yet each organ and tissues of cancers behave very differently. Similarly, research and management of STS have been evolved through histo-type specific approach, as different histological subtypes differ in clinical, biological and pathological properties.

Therefore, the study should provide data and analysis considering a subtype specific evaluation.

161: Both miR-155 and miR-203 were detectable in all samples to a different extent.

The expression values of miR155 and miR203 in each sample should be demonstrated with histo-type annotations on the same scale chart.

In addition, analysis for the clinical outcome of these micro-RNA should be done among the same histo-types; e.g., liposarcomas, even though the number is small but may able to stratify by either miR155, or miR203 alone, or combination for the prognosis.

113, Table 1

The authors analyzed the molecular biological factors associated with miR-155 and miR-203 expression in soft tissue sarcoma. There is no explanation provided to describe “the molecular biological factors” including mutation status, proteins, and mRNA.  What is the rationale to pick up these factors? From what source these factors were obtained and analyzed?

Figure 4c,

The color/line code annotation is missing

Author Response

Reviewer 2

In the manuscript, the authors report the value of miR155 and miR203 expression in soft tissue sarcoma (STS) cohort including some histo-subtypes of total 79 tumors for their prognostic potential.

The study supports the possibility of miR155 working as an oncomiR as known in some cancer types including liposarcoma, and of miR203 as a tumor suppressor. 

The concern in the study design is that the cohort includes mixed histotypes and each with only a small number so they end up treating STS as a single disease entity.

STS is a large entity of neoplasm of mesenchymal origin including more than 50 different histo-types.

Carcinoma is a large category including all neoplasm of epithelial origin, yet each organ and tissues of cancers behave very differently. Similarly, research and management of STS have been evolved through histo-type specific approach, as different histological subtypes differ in clinical, biological and pathological properties.

Therefore, the study should provide data and analysis considering a subtype specific evaluation.

Point 1: 161: Both miR-155 and miR-203 were detectable in all samples to a different extent. 

Answer: To detail this relatively broad statement, we have given the ranges of expression for miR-155 and miR-203 in chapter 2.1. (l.---)

Point 2: The expression values of miR155 and miR203 in each sample should be demonstrated with histo-type annotations on the same scale chart.

Answer: We have added a box plot visualization as Figure S1 to the revised manuscript. Additionally, we added a short paragraph in Chapter 2.1., stating: Regarding the histological subtype, especially liposarcoma and neuronal sarcoma exhibited a lower miR-155 expression in comparison to the other sarcoma entities (see Figure S1). These differences were significant (p = 0.001, Kruskal-Wallis test); however, based on relatively low case numbers (n = 7 – 21 per category).

Point 3: In addition, analysis for the clinical outcome of these micro-RNA should be done among the same histo-types; e.g., liposarcomas, even though the number is small but may able to stratify by either miR155, or miR203 alone, or combination for the prognosis.

Answer: We thank the reviewer for this suggestion. We performed survival analyses on all clustered histological subtypes with n ≥ 20. The results of univariate and multivariate Cox Regression analyses were given in S2. However, miR-155, miR-203 or the combination did not show significant effects on the survival of STS patients in liposarcoma or fibrosarcoma/NOS, but exhibited a significant association  for the expression of miR-155 or miR-203 with the survival in myogenic sarcomas (RMS + LMS; see Table S2) in a multivariate Cox regression analysis (adjusted to resection type, tumor localisation and tumor stage). We acknowledge this result in the manuscript as follows: Furthermore, then analyzing the effect of miR-155 or miR-203 on the patients survival in distinct histological subtypes, it became evident that in myogenic sarcomas (rhabdo- and leiomyosarcoma) a lowered miR-155 expression as well as a lowered miR-203 expression was significantly associated to a worsened survival of the STS patients (See Table S2).  

Point 4: 113, Table 1: The authors analyzed the molecular biological factors associated with miR-155 and miR-203 expression in soft tissue sarcoma. There is no explanation provided to describe “the molecular biological factors” including mutation status, proteins, and mRNA.  What is the rationale to pick up these factors? From what source these factors were obtained and analyzed?

Answer: The data were obtained in the respective soft tissue sarcoma cohort over a long period of time and – in the most cases – already published in the previous years. To avoid excessive self-citation, we initially decided not to include references in the manuscript; however, we see – due to the reviewers comment – that the source of these data as well as the methodology applied is necessary for the assessment of the results in the present manuscript. Therefore, we included references in Table 1. Thank you for the remark.

Point 5: Figure 4c - The color/line code annotation is missing

We fixed this.

Finally, we want to thank the reviewer for the helpful revisions, which – to our opinion – greatly helped to improve the manuscript. Thank you for your time and insightful remarks. Best regards. Thomas Greither (in behalf of the authors)

Reviewer 3 Report

The article is concise, well written and adequately supported by both the number of samples analyzed and the correlation analyses performed to determine the impact of miRNA 155 and 203 on the diagnosis, clinical features and prognosis of patients with STS. A sufficient bibliographic description is also provided.

Author Response

The article is concise, well written and adequately supported by both the number of samples analyzed and the correlation analyses performed to determine the impact of miRNA 155 and 203 on the diagnosis, clinical features and prognosis of patients with STS. A sufficient bibliographic description is also provided.

We want to thank the reviewer for the time and the kind words regarding our manuscript. Best regards. Thomas Greither (in behalf of the authors)

Reviewer 4 Report

There is a great need for the identification of new biomarkers serving as prognostic and diagnostic factors. Greither et al. presented a profound expression profile of miR-155 and miR-203, determined in 79 patients with soft-tissue sarcoma.
The strong point of this paper is data analysis and presentation. The Authors performed an extransive statistical analysis, identifying phenotype of STS patients based on miR-155 and miR-203 levels. The Authors correlated miR-155 or miR-203 expression with different molecular-pathological factors in soft tissue sarcoma.
The weak point of the study is the method of RT-qPCR data normalisation. Using only one reference gene i.e. U18 snoRNA is not sufficient, neither proper for the analysis due to different biology of snoRNA and miRNA. Authors should consider this in their future studies and find a profile of miRNAs that could serve as a reference in their studies (miRNAs with constitutive expression in STS tissue samples). However bearing in mind the scientific soundness of this study, it is worth to publish in Cancers.
Before publishing the Authors should also provide proper/full nomenclature of the tested miRNAs. It shold appear at least once in the paper. For example, in the Abstract:
Therefore, we analysed the prognostic potential of miR-155-5p (miR-155) and miR-203a-3p (miR-203) expression in a cohort of 79 STS patients.
Additionally, please check whether you analysed 203a or miR-230b:
mIR203a-3p
GUGAAAUGUUUAGGACCACUAG
miR-203b-3p
UUGAACUGUUAAGAACCACUGGA

Author Response

There is a great need for the identification of new biomarkers serving as prognostic and diagnostic factors. Greither et al. presented a profound expression profile of miR-155 and miR-203, determined in 79 patients with soft-tissue sarcoma.
The strong point of this paper is data analysis and presentation. The Authors performed an extransive statistical analysis, identifying phenotype of STS patients based on miR-155 and miR-203 levels. The Authors correlated miR-155 or miR-203 expression with different molecular-pathological factors in soft tissue sarcoma.

Point 1: The weak point of the study is the method of RT-qPCR data normalisation. Using only one reference gene i.e. U18 snoRNA is not sufficient, neither proper for the analysis due to different biology of snoRNA and miRNA. Authors should consider this in their future studies and find a profile of miRNAs that could serve as a reference in their studies (miRNAs with constitutive expression in STS tissue samples). However bearing in mind the scientific soundness of this study, it is worth to publish in Cancers.

Answer: We agree with the reviewer, that the choice of only one reference gene does not necessarily reflect the complex RNA biology of the living cell. When the experiments were performed, we also tested the microRNAs miR-325 and let-7a as putative reference genes; however, in NormFinder analyses, the U18 snoRNA expression was most stable and therefore was chosen as reference gene. Furthermore, after more literature was published demonstrating let-7a as powerful oncoMiR, and the validity of miR-325 as microRNA is still disputed, we hesitated to build a score out of these RNAs for normalization and relied on the U18 snoRNA. In addition, other snoRNAs as e.g. U48 (Han et al., Anticancer Res. 2016), U91 (pancreatic cancer/Popov et al., BMC Cancer 2015), U87 and U6 (endometrial cancer as rat model/Jurcevic et al., Cancer Cell Int 2013) have been also applied as reference genes for microRNA analyses. However, we agree that it remains still an urgent task to select valid and stable expressed reference microRNAs in soft tissue sarcoma.

Point 2: Before publishing the Authors should also provide proper/full nomenclature of the tested miRNAs. It shold appear at least once in the paper. For example, in the Abstract:
Therefore, we analysed the prognostic potential of miR-155-5p (miR-155) and miR-203a-3p (miR-203) expression in a cohort of 79 STS patients.

Answer: We fixed this point and thank the reviewer for this important clarification.

Point 3: Additionally, please check whether you analysed 203a or miR-230b:
mIR203a-3p
GUGAAAUGUUUAGGACCACUAG
miR-203b-3p
UUGAACUGUUAAGAACCACUGGA

Answer: MiR-203a was measured in this study, we fixed this.

Finally, we thank the reviewer for his or her time and the thorough review of our manuscript and the suggestions leading to these important clarifications. Best regards. Thomas Greither (in behalf of the authors)

Round 2

Reviewer 1 Report

The authors have addressed most of the concerns and left the one which are not possible at this time (as they do not have access to samples). The answers to the reviewers comments helped to understand the significance of the study.

Reviewer 2 Report

The authors provided additional information and fairly improved the revised manuscript.